# Noradrenergic projections from the locus coeruleus to the amygdala constrain fear memory reconsolidation

Josué Haubrich[1]*, Matteo Bernabo[2], Karim Nader[1]

[1]Department of Psychology, McGill University, Montreal, Canada; [2]Department of Neurology and Neurosurgery, McGill University, Montreal, Canada

**Abstract** Memory reconsolidation is a fundamental plasticity process in the brain that allows established memories to be changed or erased. However, certain boundary conditions limit the parameters under which memories can be made plastic. Strong memories do not destabilize, for instance, although why they are resilient is mostly unknown. Here, we investigated the hypothesis that specific modulatory signals shape memory formation into a state that is reconsolidation-resistant. We find that the activation of the noradrenaline-locus coeruleus system (NOR-LC) during strong fear memory encoding increases molecular mechanisms of stability at the expense of lability in the amygdala of rats. Preventing the NOR-LC from modulating strong fear encoding results in the formation of memories that can undergo reconsolidation within the amygdala and thus are vulnerable to post-reactivation interference. Thus, the memory strength boundary condition on reconsolidation is set at the time of encoding by the action of the NOR-LC.

## Introduction

New memories do not form instantly at the moment of an experience, but rather undergo a stabilization period during which they are gradually consolidated into a stable, long-term memory (*Asok et al., 2019*). Later, recall may cause the memory to return to an unstable state (i.e. destabilized) where it can be modified. Importantly, the memory must undergo a re-stabilization process called reconsolidation in order to persist (*Haubrich and Nader, 2016*). This process of destabilization-reconsolidation allows memories to adaptively change. Importantly, manipulations targeting the reconsolidation process can permanently alter a memory trace, by enhancing, impairing or modifying it, offering great treatment potential to clinicians (*Beckers and Kindt, 2017*; *Nader et al., 2013*; *Phelps and Hofmann, 2019*; *Soeter and Kindt, 2011*). For instance, one can induce the destabilization of a maladaptive memory and then block reconsolidation pharmacologically– preventing the memory from returning to a stable state and leading to memory impairment (*Nader et al., 2000*; *Suzuki et al., 2004*). Memory content can also be updated during reconsolidation, allowing it to be modified to a less aversive form (*Haubrich et al., 2015*; *Lee et al., 2017*; *Monfils et al., 2009*; *Popik et al., 2020*; *Schiller et al., 2010*). However, extreme fear learning can result in pathological memories that are resistant to change through reconsolidation (*Suzuki et al., 2004*; *Wang et al., 2009*), making them difficult to treat (*Zhang et al., 2018*).

In order, for reconsolidation interventions to work, memory destabilization must be triggered first. However, retrieval does not always induce destabilization. In rodent experiments, this is particularly the case in memories generated by high intensity fear conditioning or protocols that induce asymptotic levels of learning (*García-DeLaTorre et al., 2009*; *Gazarini et al., 2015*; *Lee, 2010*; *Morris et al., 2006*; *Winters et al., 2009*). For instance, we (*Finnie and Nader, 2020*; *Wang et al., 2009*) and others (*Holehonnur et al., 2016*) have found that while auditory fear memories created with one tone-shock pairing (1P) do undergo destabilization and can be attenuated by

*For correspondence:
josue.haubrich@mcgill.ca

**Competing interests:** The authors declare that no competing interests exist.

**eLife digest** New memories must go through a period of consolidation to become stable and long-lasting in the brain. Recalling memories can make them unstable again, so that they need reconsolidating. Treatments in which the reconsolidation process is interrupted have been used to help weaken traumatic fear memories. However, memories of severe trauma, such as in post-traumatic stress disorder, are particularly resistant to reconsolidation treatments.

Haubrich et al. used rats to study how trauma shapes memory formation and what biological mechanisms are involved in preventing the destabilization/reconsolidation cycle. The rats were exposed to a sound at the same time as receiving a mild electric shock. Half of the rats experienced the shock once, creating a 'weak' memory. The other half experienced it ten times, creating a 'strong' memory. The rats' memory of the electric shock was measured by seeing how they responded when they heard the sound again without the shock.

Some of the rats were given the drug anisomycin, an antibiotic that stops cells from making new proteins and is known for producing amnesia, to block reconsolidation of the memory after hearing the sound again. Treatment with the drug reduced future responses in the rats that had experienced the shock once, but had no effect on the rats that had experienced it ten times, demonstrating that the stronger memories were resistant to reconsolidation therapy. The rats with the strong memories also had lower levels of proteins in the brain that are involved in plasticity – the ability of the brain to change and adapt.

Haubrich et al. hypothesized that the stability of the strong memories could be caused by signaling from the locus coeruleus, a region of the brainstem involved in the response to stress. When the signaling from the locus coeruleus was blocked in the strong-memory rats, they became responsive to reconsolidation therapy with anisomycin.

These results help to better understand how traumatic memories become engrained, potentially leading to new treatment options for people with post-traumatic stress disorder.

reconsolidation blockade, a stronger, 10 tone-shock pairings (10P) protocol creates an intense fear memory that is resistant to this treatment (*Figure 1*). Thus, the strength of the memory serves as a boundary condition to determine whether memory destabilization will occur after retrieval or

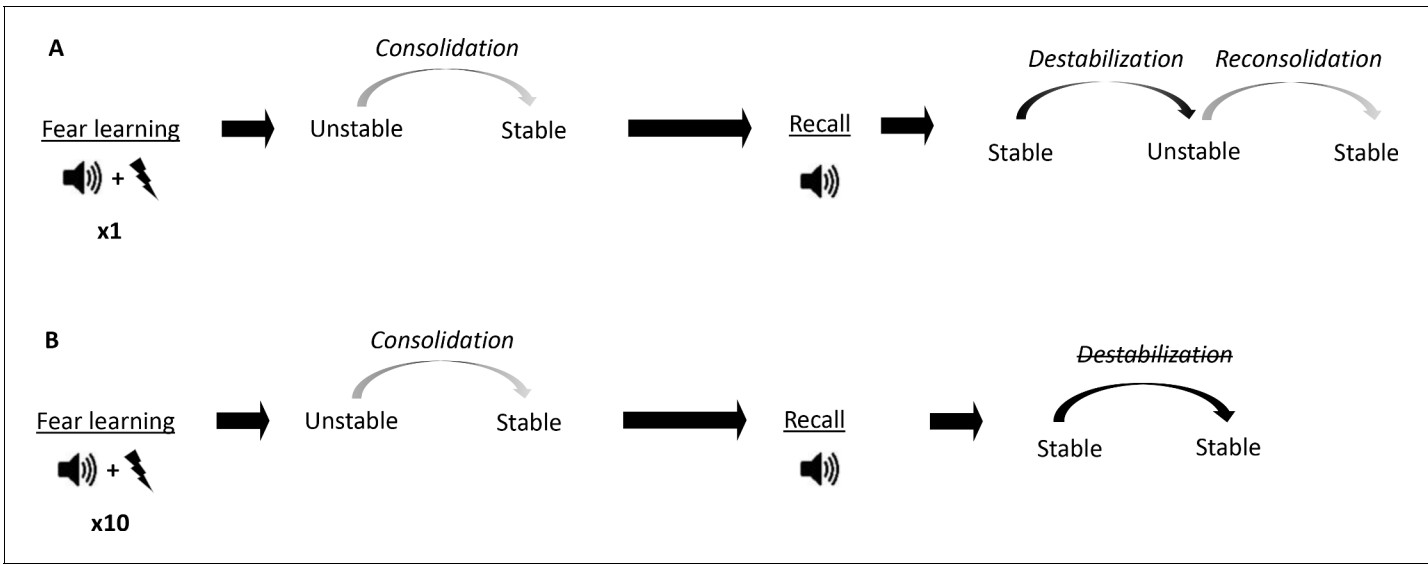

**Figure 1.** Conceptual diagram showing differences in memory processes triggered by fear conditioning of different intensities. (**A**) Auditory fear conditioning training consisting of a single tone-shock pairing (1P). The fear memory initially exists in an unstable short-term state and in order to persist as a long-term memory (LTM), it must undergo a time-dependent stabilization processes termed consolidation. Later, recall of the LTM causes memory to destabilize and become transiently unstable. Afterwards, for memory to persist it is restabilized by reconsolidation. (**B**) Auditory fear conditioning consisting of 10 tone-shock pairings (10P).

not (*Zhang et al., 2018*). This boundary condition is thought to be mediated in part by mechanisms that initialize destabilization. A well described mechanism for destabilization is GluN2B-containing NMDA receptor activation (*Ben Mamou et al., 2006*). GluN2B-NMDA receptors are reduced in reconsolidation-resistant memories (*Wang et al., 2009*) and blocking these receptors prevents destabilization (*Milton et al., 2013*). Another important mechanism involves the expression of GluA2-containing AMPA receptors, which are important for LTP maintenance (*Henley and Wilkinson, 2016*) and have been linked with memory strength and stability (*Migues et al., 2014*; *Migues et al., 2010*). In order for destabilization to occur, a transient reduction of GluA2-AMPA receptor synaptic expression is necessary (*Clem and Huganir, 2010*; *Ferrara et al., 2019*; *Hong et al., 2013*; *Rao-Ruiz et al., 2011*).

The fact that strong training leads to memories that do not reconsolidate likely reflects changes triggered at the time of memory encoding. These changes likely determine the future plasticity of the memory by altering mechanisms needed for destabilization, such as GluN2B-NMDA and GluA2-AMPA receptors expression. However, the specific links between memory encoding and reconsolidation have yet to be determined. It is well-established that emotionally arousing experiences cause increased release of noradrenaline in the amygdala (*Quirarte et al., 1998*) that modulates memory formation through the activation of β-adrenergic receptors (*McGaugh, 2000*). The locus coeruleus is the main source of noradrenergic projections in the brain (*Giustino and Maren, 2018*), and although it acts on many targets, its influence in the amygdala is critical to mediate the impact of stress on memory processes (*Giustino and Maren, 2018*; *Johansen et al., 2011*; *Kaouane et al., 2012*). For instance, overactivation of projections coming from the locus coeruleus (LC) to the amygdala have been implicated in encoding traumatic memories (*Hurlemann, 2008*) and these projections promote fear learning (*Uematsu et al., 2017*). This evidence suggests that the noradrenaline-locus coeruleus system (NOR-LC) likely contributes to the formation of maladaptive, reconsolidation-resistant fear memories. If so, NOR-LC signaling may trigger changes that determine future plasticity and destabilization, such as altered GluN2B and GluA2 expression.

Here, we compared how animals fear conditioned with a 1P or a 10P training protocol differ regarding plasticity mechanisms in the amygdala and the induction of destabilization. Using pharmacologic and chemogenetic interventions, we also tested the hypothesis that the overactivation of the NOR-LC system during severe fear encoding underlies the formation of reconsolidation-resistant memories with limited plasticity.

## Results

### R1 replicating the behavioral effects of 1 pairing vs 10 pairings fear conditioning protocol

To study the difference between mild and strong fear memories, our first aim was to assess how they differ at the behavioral level. Animals were trained in two different auditory fear conditioning tasks: to create a mild fear memory, we trained rats with one tone-shock pairing (1P), whereas strong fear memories were created using 10 tone-shock pairings (10P). The next day, fear memory was assessed in a test session where one unreinforced tone was presented (*Figure 2A*). Animals trained with 10 shocks displayed higher freezing to the tone than rats trained with one shock (Independent samples t-test t(28) = 2.308, p=0.028).

The difference in freezing during the test did not seem large due to a ceiling effect in the 10P group. To better visualize the differences in memory strength, fear-conditioned rats underwent extinction with 20 unpaired tone presentations and were tested for extinction retention in the next day (*Figure 2B*). A mixed ANOVA with training protocol as a between-subjects variable and bin as a within-subjects variable revealed a significant main effect between groups, with rats in the 10P group freezing more during the entire extinction session and at test ( $F_{1,13}$ = 58.18, p<0.001). Tukey's *post hoc* test revealed that only animals trained with 1P displayed extinction acquisition, with significant fear suppression within the extinction session (1-to-5 tone vs 16-to-20 tone: 1P group, t(52) = 3.65, p=0.02; 10P group, t(52) = 2.43, p>0.05). Also, extinction retention 24 hr later was observed only in the 1P group (1-to-5 tone vs Test: 1P group, t(52) = 3.45, p=0.03; 10P group, t(52) = 0.85, p>0.05). Therefore, in contrast with 1P, fear memories created with 10P exhibit impaired extinction learning, indicating a considerable difference in memory strength.

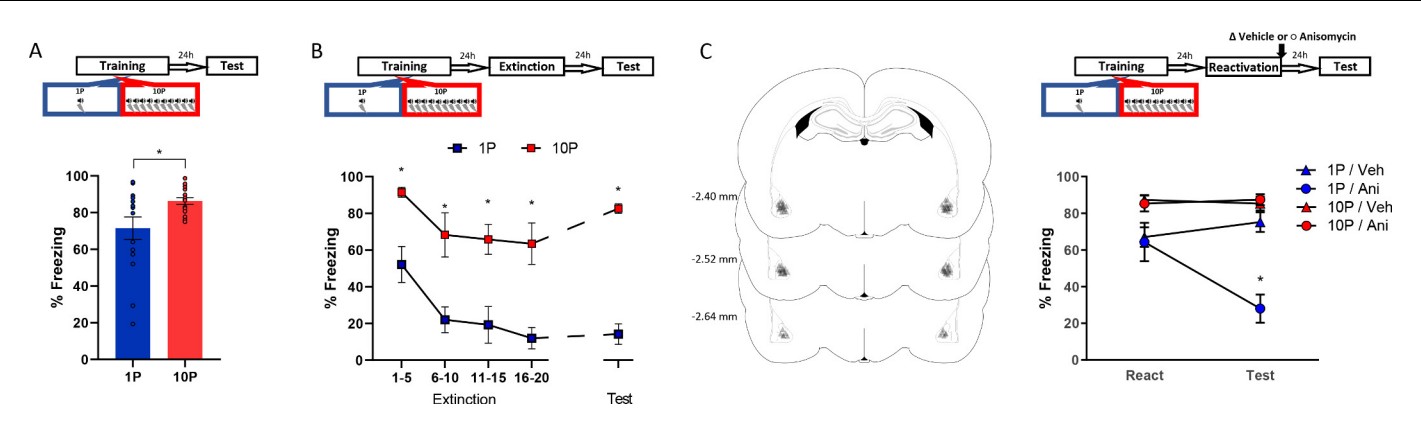

**Figure 2.** Fear conditioning training with 10 shocks creates memories that are resistant to undergo reconsolidation. Here we replicated the behavioral findings reported by *Wang et al., 2009*. Animals were trained with either one tone-shock pairing (1P) or 10 tone-shock pairings (10P) A) One day after training a retention test was conducted. Animals in the 10P group displayed higher freezing levels than those trained with a 1P (N = 15 per group). (**B**) One day after training an extinction session was conducted with 20 tone presentations, followed by a retention test the next day. Unlike in the 1P group, 10P animals did not display extinction acquisition and retention and showed higher freezing levels at all time points (N = 7/8 per group). (**C**) One day after training a reconsolidation-blockade procedure was conducted with post-reactivation infusion of anisomycin in the BLA. The next day a retention test was conducted. Left: representation of infusion sites in the BLA. Right: the reconsolidation blockade procedure was effective in disrupting fear memory only in the 1P group (N = 9/10 per group). Graphs show the mean ± s.e.m. Individual values are represented with circles. *p<0.05. The full statistics are available in the *Supplementary file 1* and individual values in *Figure 2—source data 1*.

The online version of this article includes the following source data for figure 2:

**Source data 1.** Raw data of *Figure 2*.

Next we assessed reconsolidation in 1P and 10P memories as previously described by *Wang et al., 2009*. One day after 1P or 10P training, a 1-tone test was conducted to reactivate the fear memory. The protein synthesis inhibitor anisomycin (125 µg/µl; 0.5 µl per hemisphere) was infused in the basolateral amygdala (BLA) immediately after to block reconsolidation. The effectiveness of the treatment was then evaluated in a test 1 day later. A mixed ANOVA with training and drug as between-subjects variables and session as a within-subjects variable indicated that there was a significant interaction between training, drug, and session ($F_{1,35}$ = 18.27, p<0.001). At test, post-reactivation anisomycin impaired performance in animals trained with one shock (Tukey's *post hoc* test, t(55) = 5.59, p<0.001) but had no effect in strongly trained rats (Tukey's *post hoc* test, t(55) = −0.26, p>0.05). This shows that retrieval rendered the 1P memory labile, necessitating reconsolidation shortly afterwards. On the other hand, retrieval did not render the 10P memory vulnerable to anisomycin, and hence it can be considered a reconsolidation-resistant memory.

## R2 quantification of synaptic plasticity molecules between reconsolidation-permissive vs resistant memories in the BLA

We evaluated the expression of molecules implicated with synaptic plasticity, GluN2B (*Zhang et al., 2018*) and GluA2 (*Anggono and Huganir, 2012*), between animals trained in the 1P and 10P protocols. Animals were fear conditioned in the 1P or 10P protocol, tested the next day, and their brains collected 1 hr or 24 hr later for western blot analysis of BLA tissue. Controls were kept in their home cages during the entire behavioral procedure (Home cage controls, HC). This control, although not addressing the role of shock or context alone, informs the baseline levels of the targeted proteins when no learning occurs.

A one-way ANOVA indicated significant group differences in GluN2B expression at the BLA post-synaptic density (PSD) (*Figure 3A*, left; $F_{2,11}$ = 7.34, p=0.009). The 1P group displayed an upregulation of GluN2B (Tukey's *post hoc* test, HC vs 1P: t(11) = −2.99, p=0.031), indicating that the formation of a reconsolidation-permissive memory coincides with an increase in this receptor critical for reconsolidation induction. However, 10P trained rats displayed GluN2B equivalent to HC levels (Tukey's *post hoc* test, t(11) = −0.09, p>0.05). This shows that unlike 1P memories that do reconsolidate, strong reconsolidation-resistant memories are formed without GluN2B upregulation.

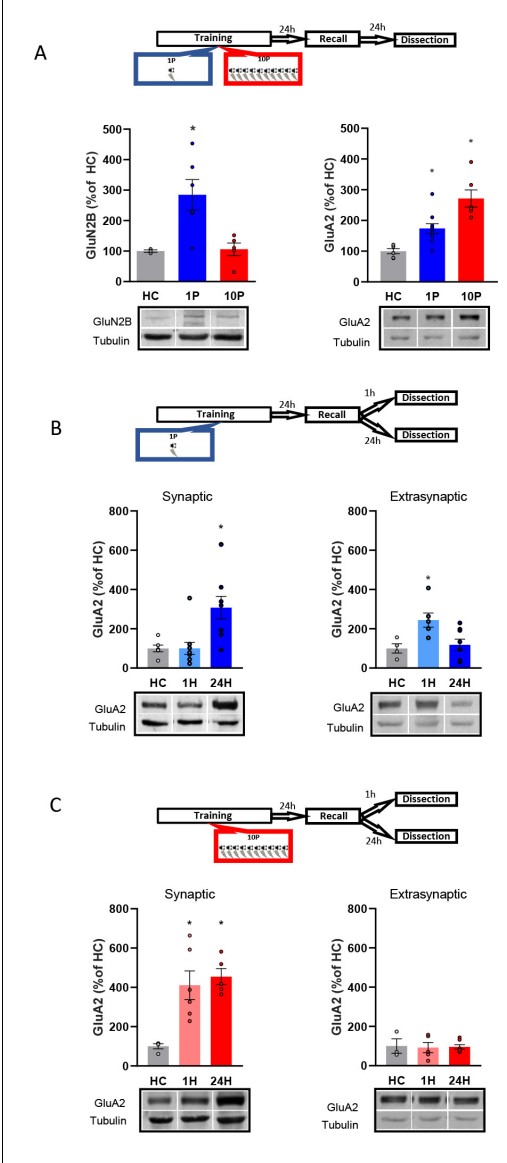

Next, we looked at how GluA2-containing AMPARs in the BLA PSD vary between 1P reconsolidation-permissive and 10P reconsolidation-resistant memories (*Figure 3A*, right). There was a significant difference between groups (One-way ANOVA, $F_{2,19}$ = 17.98, p<0.001). In comparison with HC animals, GluA2 levels of animals trained with 1P were upregulated one day after testing (Tukey's *post hoc* test, t(19) = −2.87, p=0.025), but an even higher increase was found in rats trained with 10P (Tukey's *post hoc* test, HC vs 10P: t(19) = −5.97, p<0.001; 1P vs 10P: t (19) = −3.81, p=0.003). Hence, GluA2 levels increase as a function of training strength, and the high levels found in the strong training group might reflect a resistance to memory destabilization.

Lastly, we investigated GluA2 trafficking by comparing GluA2 levels 1 hr or 24 hr after recall at both the PSD and at extrasynaptic fractions of the BLA. These time-points fall inside (1 hr) and outside (24 hr) the reconsolidation window, allowing us to capture GluA2 transient downregulation following destabilization and its levels after it has been normalized after reconsolidation (*Ferrara et al., 2019*; *Rao-Ruiz et al., 2011*). One-way ANOVA revealed a significant difference between groups (*Figure 3B*, left; $F_{2,26}$ = 7.38, p=0.003). Tukey's *post hoc* test indicated that in rats trained with 1P, synaptic GluA2 levels were increased 24 hr after recall but were downregulated to HC levels shortly after recall (HC vs 1 hr: t(24) = 0.002, p>0.05; HC vs 24 hr: t(24) = 2.97, p=0.018; 1 hr vs 24 hr: t(24) = 3.45, p=0.006). GluA2 levels also differed between groups at the extrasynaptic fractions (*Figure 3B*, right; One-way ANOVA, $F_{2,17}$ = 5.74, p=0.014), with higher GluA2 expression 1 hr after recall compared to HC animals (Tukey's *post hoc* test, HC vs 1 hr: t(15) = −2.85, p=0.031; HC vs 24 hr, t (15) = −0.38, p>0.05; 1 hr vs 24 hr, t(15) = 2.97, p=0.024). This indicates that shortly after recall GluA2 is transiently removed from the synapse, rendering the 1P memory labile. This pattern of GluA2 trafficking, however, was not observed in 10P reconsolidation-resistant animals. GluA2 levels were unchanged by recall at both the synaptic (*Figure 3C*, left; One-way ANOVA, $F_{2,21}$ = 5.46, p=0.012; Tukey's *post hoc* test, HC vs 1 hr: t(21) = −2.95, p=0.02; HC vs 24 hr: t(21) = −2.96; p=0.02, 1 hr vs 24 hr: t(21) = −0.01, p>0.05), and extrasynaptic fractions (*Figure 3C*, right; One-way ANOVA, $F_{2,12}$ = 0.024, p>0.05). This shows that strong memories that fails to become labile upon recall display increased GluA2 synaptic expression.

**Figure 3.** Reconsolidation-resistant memories created with 10P display reduced plasticity mechanisms in comparison to memories that are reconsolidation-permissive. (**A**) Animals were trained with either 1P or 10P and were tested the next day. One day later BLA samples were collected and the postsynaptic levels of GluN2B and GluA2 quantified. Left: in comparison with HC controls, GluN2B increases in the 1P group but not in the 10P group (N = 3/6 per group). Right: 1P rats displayed higher GluA2 levels than HC, and 10P pairing displayed higher GluA2 levels than all other groups (N = 6/10 per group). (**B**) Animals were trained in the 1P protocol and were tested the next day. BLA samples were collected either 1 hr or 1 day after test to probe for postsynaptic and extrasynaptic GluA2 expression. Left: postsynaptic GluA2 is at HC levels 1 hr after test and is increased 24 hr after test (N = 6/11 per group). Right: extrasynaptic GluA2 is increased 1 hr after test and at HC levels 24 hr after test (N = 4/8 per group).
*Figure 3 continued on next page*

*Figure 3 continued*

(C) Animals were trained in the 10P protocol and were tested the next day. BLA samples were collected either 1 hr or 1 day after test to probe for postsynaptic and extrasynaptic GluA2 expression. Left: postsynaptic GluA2 is increased in comparison to HC at both 1 hr and 24 hr after test (N = 4/6 per group). Right: Extrasynaptic GluA2 is at HC levels at both 1 hr and 24 hr after test (N = 3/7 per group). Graphs show the mean ± s.e.m. Individual values are represented with circles. *p<0.05. The full statistics are available in the *Supplementary file 2* and individual values in *Figure 3—source data 1*.

The online version of this article includes the following source data for figure 3:

**Source data 1.** Raw data of *Figure 3*.

Overall, this set of data shows that, unlike reconsolidation-permissive 1P memories, memories created by 10P are reconsolidation-resistant with have attenuated plasticity, explaining why reconsolidation-blockade is ineffective in strong memories.

## R3 B-adrenergic receptors activation during 10P training shifts memories into a reconsolidation-resistant state

Here, we investigated whether β-adrenergic receptor activation during strong training plays a role in the expression of pro-plasticity molecules. After showing that strong training results in memories with decreased GluN2B and increased GluA2 postsynaptic expression in the BLA, we investigated if the β-adrenergic inhibitor propranolol would affect such outcomes. Animals were injected with either propranolol (10 mg/mL/kg, i.p.) or vehicle and 20 min later were trained in the 10P fear conditioning protocol. Rats were tested one day later and sacrificed 24 hr post-test when their brains were collected for western blot analysis. In comparison with the vehicle groups, propranolol treatment increased GluN2B (*Figure 4B*, left; Independent samples t-test t(17) = 2.47, p=0.025) and decreased GluA2 (*Figure 4A*, right; Independent samples t-test t(9) = 2.41, p=0.039) in the BLA. Therefore, blocking β-adrenergic signaling during 10P training modulates GluN2B and GluA2 toward the levels found in 1P memories that do reconsolidate. Importantly, no direct comparison with the 1P and HC groups were made. Thus, in respect to 1P memories and HC levels, the extent to which propranolol before strong training modulates GluN2B and GluA2 remains to be established.

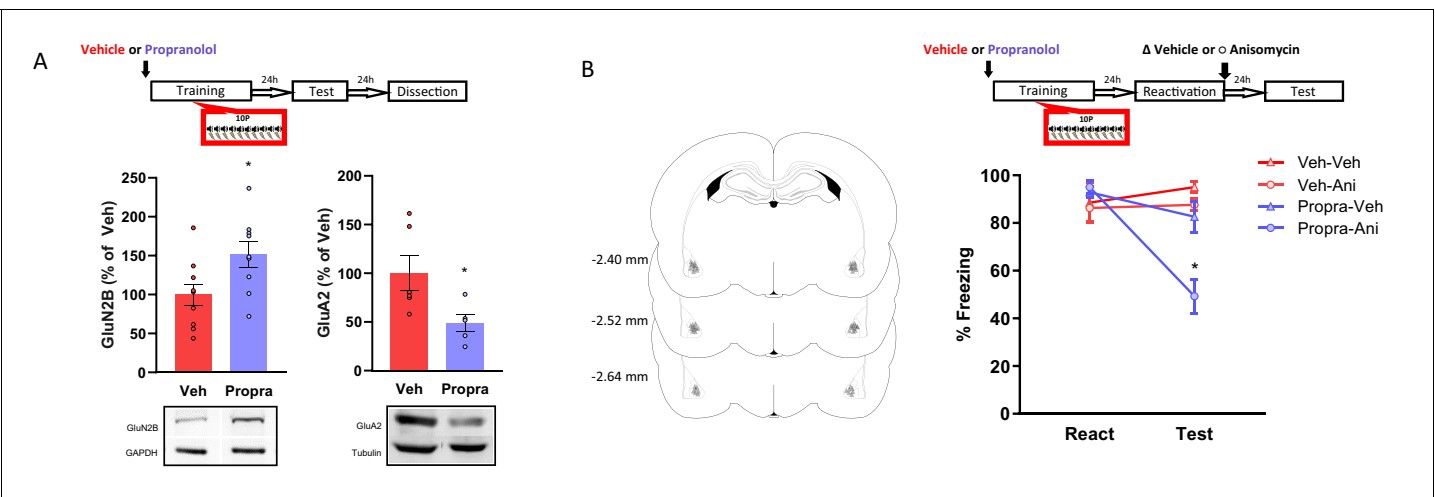

**Figure 4.** The activation of β-adrenergic receptors is necessary memory to shift from a reconsolidation-permissive to a reconsolidation-resistant state. Animals received an injection of propranolol (i.p. 10 mg/kg) or saline and 15 min later were trained in the strong training protocol. (A) One day after training rats were tested, and in the next day their BLA was collected to probe for postsynaptic GluN2B and GluA2. Left: propranolol treatment upregulated GluN2B (N = 9/10 per group). Right: propranolol treatment downregulated GluA2 (N = 5/6 per group). (B) One day after training a reconsolidation-blockade procedure was conducted with intra-BLA post-reactivation anisomycin. The next day a retention test was conducted. Left: representation of infusion sites in the BLA. Right: the reconsolidation blockade procedure was effective in disrupting fear memory only in animals receiving propranolol before training (N = 7/8 per group). Graphs show the mean ± s.e.m. Individual values are represented with circles. *p<0.05. The full statistics are available in the *Supplementary file 3* and individual values in *Figure 4—source data 1*.

The online version of this article includes the following source data for figure 4:

**Source data 1.** Raw data of *Figure 4*.

Next, we assessed if blocking β-adrenergic signaling during strong training would affect a memory's susceptibility to undergo reconsolidation later on. Again, animals were injected with propranolol or vehicle and trained in a 10P fear conditioning protocol. Afterwards, reconsolidation blockade was conducted, and its effectiveness evaluated as described above. A three-way ANOVA indicated a significant interaction between treatment 1, treatment two and session (*Figure 4B*; $F_{1,54}$ = 4.14, p=0.047). During reactivation, animals that received vehicle and propranolol displayed the same freezing levels, indicating that propranolol did not impair strong fear learning (Tukey's *post hoc* test, t(54) = −0.57, p>0.05). In vehicle-injected animals, post-reactivation anisomycin infusion had no effect on fear expression at the test (Tukey's *post hoc* test, veh + veh vs veh + anisomycin: t(54) = 0.98, p>0.05). Thus, the strong fear memory was reconsolidation-resistant. Pre-training propranolol treatment, however, rendered animals susceptible to post-reactivation anisomycin, resulting in amnesia at test (Tukey's *post hoc* test, propranolol + veh vs propranolol + anisomycin: t(54) = 4.56, p<0.001).

These data reveal that reconsolidation-resistance induced by 10P training requires the activation of β-adrenergic receptors during initial learning.

## R4 projections from the Locus Coeruleus to the BLA during 10P training are critical for memories to be formed in a reconsolidation resistant state

We hypothesized that the LC-to-BLA projection could mediate the formation of reconsolidation-resistant memories in 10P training. First, we quantified the expression of the activity-regulated gene c-fos in the LC 90 min after 1P and 10P training (*Figure 5A*). We observed that 10P training resulted in higher c-fos activity (*Figure 5A*; Independent samples t-test t(4) = 4.35, p=0.01), indicating that the LC is overly engaged during the formation of strong fear memories. Next, we used a chemogenetic approach to silence LC to BLA projections during strong training (*Figure 5B–C*). Virus (pAAV-hSYN-DIO-hM4Di-mCHerry) expressing the inhibitory DREADD receptor hM4Di was infused in the LC. Controls were infused with viruses not expressing hM4Di (pAAV-hSYN-tdTomato). After 3 months, there was robust somatic expression in the LC and terminal expression in the BLA

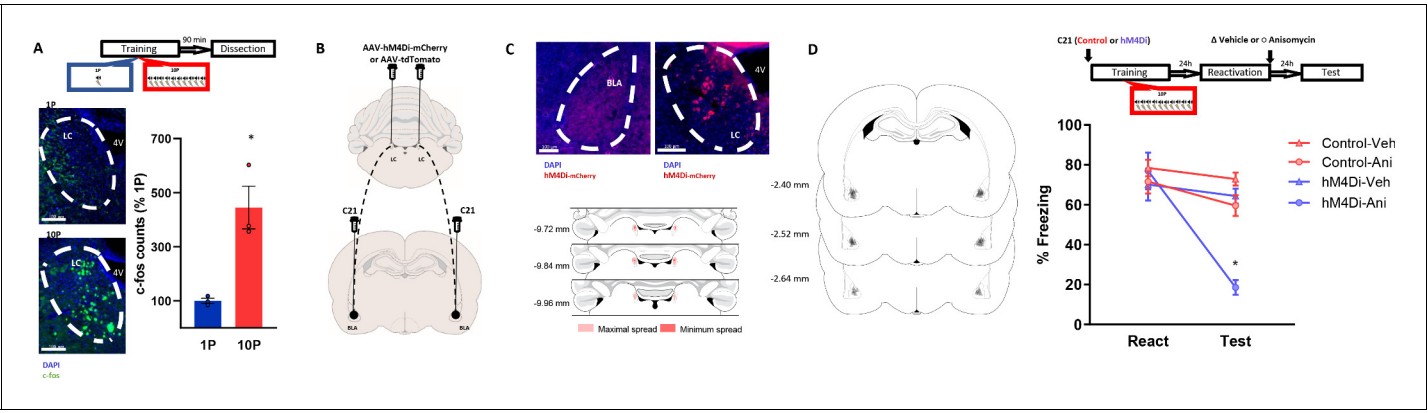

**Figure 5.** Projections from the LC to the BLA are necessary for the formation of reconsolidation resistant memories. (A) Animals underwent 1P or 10P fear conditioning training, were sacrificed 90 min later and slices were collected to verify c-fos expression. Training resulted in greater LC c-fos expression in the 10P group than in the 1P group (N = 3 per group). (B) Animals were infused in the LC with pAAV-hM4Di-mCHerry or the control pAAV-tdTomato. Terminal projections in the BLA were silenced with local infusion of the DREADD agonist C21 (2 μg/μL, 0.5 uL/side). (C) Top: after 3 months from the viral injections, it was observed robust somatic expression in locus coeruleus neurons (right) and terminal expression in the BLA (left). Bottom: Light red represents the maximal, and dark red the minimum viral expression spread observed in the LC included in the analysis. (D) After 3 months from viral infusions, animals were infused in the BLA with C21 to block LC-BLA projections and were and trained in the 10P training protocol 5 min later. One day after training a reconsolidation-blockade procedure was conducted with intra-BLA post-reactivation anisomycin. The next day a retention test was conducted. Left: representation of infusion sites in the BLA. Right: the reconsolidation blockade procedure was effective in disrupting fear memory only in animals infused with the hM4Di virus (N = 7/9 per group). Graphs show the mean ± s.e.m. Individual values are represented with circles. *p<0.05. The full statistics are available in the *Supplementary file 4* and individual values in *Figure 5—source data 1*.

The online version of this article includes the following source data for figure 5:

**Source data 1.** Raw data of *Figure 5*.

(*Figure 5C*). To silence LC inputs to the BLA during strong fear learning, we infused the DREADD agonist C21 in the BLA 5 min before 10P training. One day later, rats were infused in the BLA with anisomycin or vehicle after retrieval to determine if reconsolidation could occur (*Figure 5D*). There was a significant interaction between DREADD, drug, and session (Three-way ANOVA, $F_{1,54} = 8.26$, p=0.006). During reactivation, all groups exhibited similar freezing behavior indicating that silencing LC-BLA projections did not impair strong fear learning (Tukey's *post hoc* test, tdTomato vs hM4Di: t (54) = 1.05, p>0.05). Control animals infused with anisomycin did not show a memory impairment on a subsequent test (Tukey's *post hoc* test, tdTomato + veh vs tdTomato + anisomycin: t(54) = 1.68, p>0.05), indicating that reconsolidation did not occur in these animals. When the LC to BLA projections were silenced during strong memory formation, however, recall did enable reconsolidation since anisomycin caused amnesia in these animals (Tukey's *post hoc* test, hM4Di + veh vs hM4Di + anisomycin: t(54) = 5.64, p<0.001).

These results show that inputs from the LC to the BLA during fear learning promote memory formation into a fixed state that is resistant to change. If these projections are silenced, even the memory of a strong fear experience can become labile upon recall and be modulated through reconsolidation.

## Discussion

Previous studies revealed that unlike mild fear memories, very strong ones do not destabilize upon recall (*Holehonnur et al., 2016*; *Wang et al., 2009*). Some of the mechanisms required to induce destabilization have been identified (*Haubrich and Nader, 2016*; *Zhang et al., 2018*), but not much is known regarding how severe fear experiences end up forming reconsolidation-resistant memories. In the current study we investigated the hypothesis that strong memories are different than mild ones due to the action of the NOR-LC system on encoding. Our results show that strong fear conditioning involves elevated β-adrenergic receptor activation which modulates encoding to produce memories that do not destabilize. At the systems level, reconsolidation-resistant memory formation entails inputs from the LC to the BLA during strong fear learning, and if these projections are silenced, even the memory of a strong fear experience can later undergo destabilization and reconsolidation.

This work builds on previous studies showing that the induction of reconsolidation involves the presence of specific cellular mechanisms in the amygdala, such as the activation of GluN2B-containing NMDA receptors (*Wang et al., 2009*) and transient reduction of GluA2-containing AMPA receptors upon retrieval (*Rao-Ruiz et al., 2011*). Here, we first replicated the seminal findings of *Wang et al., 2009* by showing that, while memories created with one tone-shock pairing destabilize upon recall, memories created with 10 pairings do not. Also, memories created with 10 pairings are much more robust and display impaired extinction learning, a hallmark of posttraumatic stress disorder (*Pitman et al., 2012*). When looking at GluN2B expression, we confirmed previous reports showing that strongly-trained animals display lower GluN2B levels than mildly trained ones (*Holehonnur et al., 2016*; *Wang et al., 2009*). However, here we included a home cage control group in our analysis which revealed that GluN2B is not actually downregulated by strong training. In fact, GluN2B is normally upregulated by fear learning, which is hampered when the experience is very intense. We also examined GluA2 expression, which is central for synaptic stability and must be endocytosed after recall for memory destabilization (*Hong et al., 2013*; *Rao-Ruiz et al., 2011*). By comparing GluA2 levels of trained animals with home cages, we found a moderate increase in rats trained with one pairing and a significantly higher increase in rats trained with 10 pairings, indicating increased stability of strong memories. Retrieval transiently reduced synaptic GluA2 to home cage levels only in the mild memory, further indicating that the strong memory was overly stable and unable to destabilize. Overall, these data show that strong memories are stored in a state with different metaplastic properties that makes them resistant to reconsolidation.

Most studies concerning the boundary conditions of reconsolidation focus on the mechanisms implicated at the time of retrieval, while those at initial encoding have not received much attention. Here we show that conditions during encoding are crucial to defining the memory's future plasticity. Emotionally arousing experiences cause increased release of noradrenaline from the locus coeruleus to the amygdala (*Quirarte et al., 1998*), which increases its activation (*Rodríguez-Ortega et al., 2017*) and promotes fear learning (*Gazarini et al., 2013*; *Uematsu et al., 2017*). Moreover, the

overactivation of this system has been implicated with traumatic memory formation in humans (*Hendrickson and Raskind, 2016*; *Rombold et al., 2016*). To investigate the role of noradrenaline signaling on the formation of reconsolidation resistant memories, we used pharmacologic and chemogenetic approaches. First, we pharmacologically blocked β-adrenergic receptors during strong fear learning and re-assessed plasticity mechanism in the BLA and reconsolidation induction. Propranolol before strong training modulated GluN2B and GluA2 expression towards levels seen in weakly trained animals— increasing GluN2B and decreasing GluA2 synaptic expression. Likewise, these strong memories formed under β-adrenergic receptors blockade can be destabilized following retrieval. Next we investigated the role of projections from the LC to BLA during strong fear memory formation. To this end, LC neurons were infected with the inhibitory DREADD receptor HM4Di and its projections to the BLA were silenced during strong training with the local infusion of the DREADD agonist C21. When these specific LC to BLA projections were silenced during strong training, recall was successful in triggering destabilization. As a result, infusion of a protein synthesis inhibitor blocked reconsolidation, leading to amnesia. Importantly, both pharmacologic and chemogenetic manipulations of noradrenaline signalling did not alter rats' freezing compared to control animals. This suggests that noradrenaline blockade did not reduce memory strength per se. Overall, this reveals that the activation of LC-BLA pathway during severe fear learning is necessary for memories to encode in a fixed state and do not destabilize in the future (*Figure 6*). If this pathway is silenced, however, even the memory of a very aversive event displays plasticity and can undergo destabilization upon recall.

Reconsolidation is a critical biological feature to maintain memory's relevance to guide future behavior (*Lee, 2009*; *Lee et al., 2017*). Further, reconsolidation-targeted treatments provide a unique opportunity to weaken pathological fear memories (*Monfils and Holmes, 2018*; *Phelps and Hofmann, 2019*). Nonetheless, the intensity of the experience is an important parameter which can modulate whether reconsolidation can happen in the future, curbing the efficacy of some treatments. Thus, in order to better understand and treat disorders implicated in severe fear, it is crucial to elucidate how mild and strong memories differ. The current work underscores the widely accepted— but understudied— possibility that mild and strong fear memories are neurobiologically distinct. We have described how the NOR-LC system shapes memory encoding towards a maladaptive state that is resistant to change. This is achieved by triggering molecular modifications that increase memory stability at the expense of plasticity. The identification of this mechanism advances our understanding of the boundary conditions to reconsolidation. Moreover, this knowledge will help guide future research to understand why strong memories are so implacable, and how resistance to treatment can be overcome.

## Materials and methods

### Rats

Male Sprague Dawley rats (275–300 g at arrival; Charles River, Quebec, Canada) were housed in pairs in plastic cages in a temperature-controlled environment (21–23°C) with ad libitum access to food and water and maintained on a 12 hr light/dark cycle (lights on at 7:00 A.M.). All experiments were conducted during the light phase. In all experiments animals were randomly assigned to each behavioral condition. Sample size estimates were determined based on effect sizes observed in previous reports using similar assays (*Hardt et al., 2010*; *Holehonnur et al., 2016*; *Wang et al., 2009*) resulting in statistical power estimates between 70% and 90%. Each rat was handled for at least 4 days before the behavioral procedures. All procedures were approved by McGill's Animal Care Committee (Animal Use Protocol #2000–4512) and complied with the Canadian Council on Animal Care guidelines.

### Surgery

Animals were anesthetized with a mixture of ketamine (50 mg/ml), xylazine (3 mg/ml), and Dexdomitor (0.175 mg/ml) injected intraperitoneally. Analgesic treatment was administered subcutaneously before surgery (carprofen; 5 mg/ml). Stainless-steel 22-gauge cannulae were bilaterally implanted in the basolateral amygdala (AP, −3.0 from Bregma; ML, + / − 5.1 from the midline; DV, −8.0 from the skull surface). The cannulae were kept in place by dental cement tightly fixed to the skull with three

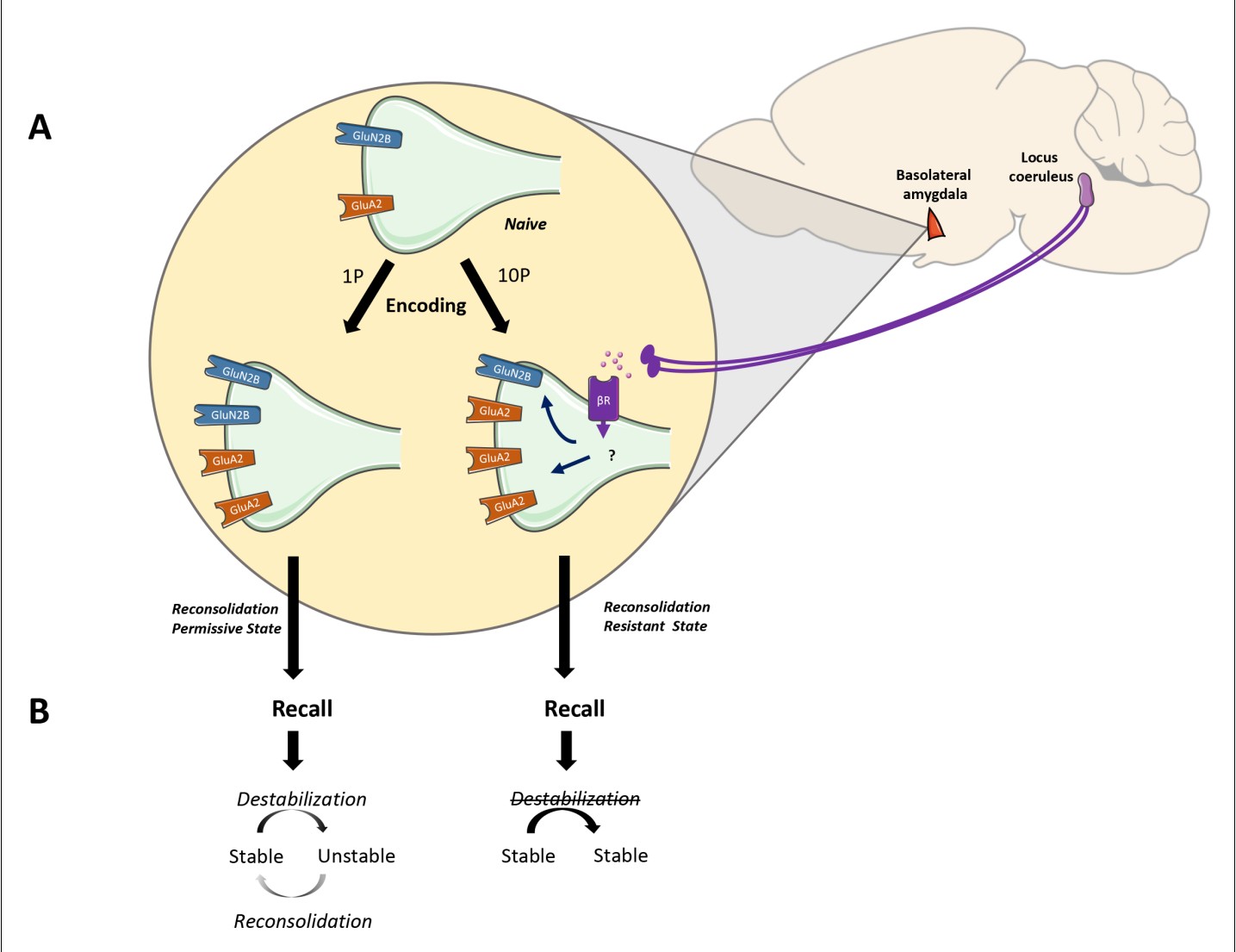

**Figure 6.** The noradrenaline-locus coeruleus system drives memory formation towards a reconsolidation-resistant state. (**A**) Fear learning engages plasticity in the amygdala that promotes long-term memory storage. Mildly aversive fear learning (1P) results in the formation of a memory where both GluN2Band GluA2 receptor subunits are upregulated. Extreme fear learning (10P) overactivates the noradrenaline-locus coeruleus systems, which prevents GluN2B upregulation and boosts GluA2 expression. (**B**) Upon recall, memory created with 1P fear conditioning is destabilized and undergoes reconsolidation. The memory formed under the ND-LC modulation does not.

stainless-steel screws. Obturators were then inserted into the cannulae to prevent blockage. An intraperitoneal injection of Antisedan (0.5 mg/ml) was administered after surgery to reverse the anesthesia, and the animals were placed in individual cages on heating pads until they woke up. The rats were then monitored daily during a 1 week recovery period, before the beginning of the behavioral procedures.

For viral injections in the LC, animals were anesthetized as above and a 33-gauge stainless steel injector was inserted into the brain (coordinates, from bregma: A/P −9.8 mm, L ± 1.3 mm, D/V −7.5 mm from the skull surface). The injector was attached by polyethylene tubing (Intramedic #427406) to 10 uL Hamilton syringes and driven at 0.4 μL/min by a microinfusion pump (KD Scientific; model 780220). A volume of 0.7 uL/side of pAAV-hSYN-DIO-hM4D(Gi)-mCHerry ($1.5 \times 10^{13}$ GC/mL, Neurophotonics Centre - ULAVAL) or pAAV-hSYN-tdTomato ($1.5 \times 10^{12}$ GC/mL, Neurophotonics Centre - ULAVAL) was infused at a rate of 0.05 uL/min and the injector was withdrawn 15 min afterwards. To target LC terminals in the BLA, 7 weeks after viral infusions stainless steel cannulae were

bilaterally implanted in the BLA as described above but using the following coordinates: A/P −3.0 mm, L ± 5.1 mm, D/V −9.0 mm from bregma.

## Drugs and drug delivery

Anisomycin (125 µg/µl; Sigma-Aldrich) was dissolved in equimolar HCl and sterile saline. Propranolol (10 mg/mL; Sigma-Aldrich) was dissolved in sterile saline. C21 (2 µg/µL, Hello Bio) was dissolved in sterile water. The pH-value of each solution was adjusted to 7.2–7.5. Propranolol was administered intraperitoneally at a volume of 1 mL/kg 15 min before training. Anisomycin (immediately after reactivation) and C21 (5 min before training) were infused bilaterally into the amygdala using a 23 gauge injectors connected to Hamilton syringes via 20 gauge plastic tubes. A total volume of 0.5 µl per side was infused by a microinfusion pump at a rate of 0.125 µl/min. Injectors were left in place for an additional minute to ensure proper drug diffusion.

## Auditory fear conditioning task

### Habituation

Rats were habituated to Context A for 15 min on Day 1 and Day 2. Context A was brightly lit and consisted of Plexiglas conditioning boxes enclosed in soundproof chambers (30.5 × 24.1 × 21.0 cm; Med Associates). The boxes each had black and white-striped front and back walls and rounded white plastic walls on the sides, as well as a white plastic floor. Peppermint-scented water was sprayed on the floor and the walls of the boxes.

### Training

On Day 3, the rats were trained in Context B, where they received either 1 or 10 pairings of a tone (20 s, 5 kHz, 75 dB) coterminating with a footshock (1 s, 1.0 mA). The first tone was presented 2 min after the rats were placed inside the boxes. The interpairing interval was of 60 s. After the last pairing, rats remained in the boxes for another 60 s before being returned to their home cages. For the rats to differentiate between the training and testing contexts, Context B consisted of a dimly lit room with transparent Plexiglas conditioning boxes, also enclosed in soundproofed chambers (30 × 25 × 30 cm; Coulbourn Instruments). A stainless-steel grid floor provided the shocks. A fan was on as background noise, and vanilla-scented water was sprayed on the boxes' walls.

### Reactivation and test

Reactivation took place in context A one day after training and entailed one 30 s tone presentation without footshock after an initial 2 min acclimation. Rats remained in the boxes for 60 s after the tone presentation. Test session was identical to the reactivation session and was always conducted 1 day after either training, reactivation or extinction sessions, depending of the experimental design.

### Extinction

One day after training, rats received 20 presentations of the tones (30 s each) without any footshocks in context A. The intertone interval was of 60 s. Rats remained in the boxes for 60 s after the last tone presentation.

Digital cameras recorded the animals' behavior, and memory was evaluated by a blind experimenter measuring the time spent freezing during the tone presentation, using Freeze View software (Actimetrics). Freezing was defined as immobilization except for respiration.

## Western blots

For experiments requiring Western blots, rats were anesthetized with isoflurane either 1 hr or 24 hr after test and were decapitated, and their brains were removed and frozen at −80°C until further use. The basolateral amygdala was dissected from each frozen brain in the cryostat using a neuro punch (1 mm; Fine Science Tools) and homogenized in ice-cold Tris-HCl buffer (30 mm, pH 7.4) containing 4 mm EDTA, 1 mm EGTA, and a protease inhibitor cocktail (Complete; Roche). The subcellular fractionation procedure performed was described previously (*Migues et al., 2010*). Briefly, the amygdala homogenates were centrifuged at 3,000 × g for 10 min at 4°C to remove the nuclei. The supernatant was then centrifuged at 100,000 × g (Beckman Coulter) for 1 hr at 4°C. The pellets were resuspended in 50 µl of 0.5% Triton X-100 homogenization buffer and incubated for 20 min on ice,

before being layered over 100 µl of 1 M sucrose solution and centrifuged at 100,000 × g for 1 hr at 4˚C. The layer remaining above the sucrose, which contained the extrasynaptic receptors, was collected, and the Triton X-100-insoluble material that sedimented through the sucrose layer, containing the postsynaptic densities, was resuspended in 40 µl homogenization buffer and stored at −80˚ C. Total protein concentration was determined by the BCA protein assay kit (Pierce).

Western blots were performed with 8% SDS-PAGE. The proteins were then transferred overnight onto nitrocellulose membranes. The membranes were incubated for 1 hr in blocking solution [0.1% Tween 20% and 5% BSA in Tris-buffered saline (TBS)], rinsed with TBS, and then incubated with polyclonal GluN2B (1:250; 71–8600 Thermo Fischer), GluA2 (1:2000; AB10529 Millipore), GAPDH (1:10,000; 6C5 Abcam) or β-Tubulin (1:10,000; T 8328Sigma) overnight. After TBS washes, the membranes were then incubated for 1 hr with a secondary antibody (goat anti-rabbit horseradish peroxidase-linked IgG [NA934V GE Healthcare] or sheep anti-mouse horseradish peroxidase-linked IgG [NA931V GE Healthcare]) and revealed with the Pierce ECL 2 Western Blotting Substrate (Thermo Fischer). The membranes were then scanned on a Storm Laser scanner (Molecular Dynamics) and the signals quantified using image analysis software (ImageLab, BioRad). The raw values obtained were normalized to the loading control values and expressed as a percentage of the control group.

## Immunohistochemistry

To analyze c-fos expression, coronal brain slices were incubated for 1 hr in blocking solution at room temperature (3% NGS, 0.3% Triton X-100) and then for 20 hr with anti-c-fos primary rabbit antibody (1:500, 226.003; Synaptic Systems, Göttingen, Germany) for 24 hr. Sections were washed and incubated with anti-rabbit Alexa-488 secondary antibody (1:500, Jackson Immunoresearch, West Grove, PA) for 2 hr at room temperature. Afterwards, sections were washed, mounted on slides and immediately coverslipped with Fluoromount-G with DAPI (Thermo Fischer). Images were examined by fluorescence microscopy (Leica DM 5000 B) and c-fos positive cells were counted bilaterally from two LC coronal slices for each animal with ImageJ.

## Histology

To identify cannulae placements, brains were removed and post-fixed in 10% formalin-saline, 20% sucrose solution and cryo-sectioned at 50 µm thickness. The slides were examined by bright-field light microscopy (Olympus IX81) by an experimenter blind to the group assignments. Only animals with injector tips bilaterally positioned within the BLA were included in the data analysis. Rats with extensive damage were excluded from analysis. To verify viral expression, brains were post-fixed in paraformaldehyde 4%, phosphate buffer 0.2M for 3 days followed by 10% formalin-saline, 20% sucrose solution for another 3 days and were then cryo-sectioned at 80 µm thickness. Slices were mounted onto slides with Fluoromount-G, with DAPI (Thermo Fischer) and examined by fluorescence microscopy (Leica DM 5000 B). Only animals with somatic viral expression in the LC were included in the data analysis, and rats with extensive damage were excluded. Across all experiments, a total of 32 animals with cannula placements or viral expression outside the targeted areas, blocked cannulae, and/or with tissue damage were excluded from analysis.

## Statistics

We used two-tailed independent-samples t test, one-way, two-way and three-way independent, repeated measures and mixed ANOVA for data analysis. Tukey's p*ost hoc* tests were further used to identify the critical differences that contributed to significant interaction. Type-one error rate was set at 0.05.

# Additional information

## Funding

| Funder | Grant reference number | Author |
| --- | --- | --- |
| Natural Sciences and Engineering Research Council of Canada | 203523 | Karim Nader |

| Canadian Institutes of Health Research | 238757 | Karim Nader |

The funders had no role in study design, data collection and interpretation, or the decision to submit the work for publication.

## Author contributions

Josué Haubrich, Conceptualization, Investigation, Visualization, Methodology, Writing - original draft, Writing - review and editing; Matteo Bernabo, Investigation, Methodology, Writing - review and editing; Karim Nader, Conceptualization, Supervision, Funding acquisition, Writing - review and editing

## Author ORCIDs

Josué Haubrich (iD) https://orcid.org/0000-0002-3632-5566

## Ethics

Animal experimentation: All procedures were approved by McGill's Animal Care Committee (Animal Use Protocol #2000-4512) and complied with the Canadian Council on Animal Care guidelines.

## Decision letter and Author response

Decision letter https://doi.org/10.7554/eLife.57010.sa1
Author response https://doi.org/10.7554/eLife.57010.sa2

# Additional files

## Supplementary files

- Supplementary file 1. Statistical analysis of the experiments reported in *Figure 2*.

- Supplementary file 2. Statistical analysis of the experiments reported in *Figure 3*.

- Supplementary file 3. Statistical analysis of the experiments reported in *Figure 4*.

- Supplementary file 4. Statistical analysis of the experiments reported in *Figure 5*.

- Transparent reporting form

## Data availability

All data is available via Dryad https://doi.org/10.5061/dryad.70rxwdbtq.

The following dataset was generated:

| Author(s) | Year | Dataset title | Dataset URL | Database and Identifier |
|-----------|------|---------------|-------------|-------------------------|
| Haubrich J, Bernabo M, Nader K | 2020 | Noradrenergic projections from the locus coeruleus to the amygdala constrain fear memory reconsolidation | http://dx.doi.org/10.5061/dryad.70rxwdbtq | Dryad Digital Repository, 10.5061/dryad.70rxwdbtq |

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
