## [Decision Letter]

**Acceptance summary:**

Memory retrieval is often followed by a process of memory reconsolidating, which renders such memories labile and potentially subject to disruption or updating. Strong memories, however, are resistant to reconsolidating. The present paper uncovers the neural mechanisms that regulate the switch from reconsolidation-resistant to reconsolidating-prone memories.

**Decision letter after peer review:**

Thank you for submitting your article "Noradrenergic projections from the locus coeruleus to the amygdala constrain fear memory reconsolidation" for consideration by *eLife*. Your article has been reviewed by three peer reviewers, including Mihaela D Iordanova as the Reviewing Editor and Reviewer #1, and the evaluation has been overseen by Kate Wassum as the Senior Editor. The following individual involved in review of your submission has agreed to reveal their identity:); Ingrid Reverte (Reviewer #2).

The reviewers have discussed the reviews with one another and the Reviewing Editor has drafted this decision to help you prepare a revised submission.

Summary:

The authors present an exciting manuscript studying the molecular and circuit mechanisms of fear memories that are reconsolidation resistant. The procedure used a 1-trial vs. a 10-trial fear conditioning design and led to a memory that is either susceptible or resistant to post-reactivation interference and to a different molecular signature (GluN2B and GluA2 glutamate subunits) in the basolateral amygdala after recall. The authors further our understanding on the resiliency to reconsolidation by exploring the role of β-adrenergic signaling during the formation of such memories. Through a series of experiments, the authors reveal the importance of b-adrenergic blockage and the role of LC to BLA projections in transitioning such strong reconsolidation-resistant memories to reconsolidation-prone memories.

This is an excellent analysis of the physiological characteristics of stable strong fear memories which are normally not subject to reconsolidation. This work is important because it specifies the physiological regulation of memories that are resistant to change. As a result this work carries important implication for other research that studies memory modification and carries clear clinical relevance.

Essential revisions:

1) Please report all statistics (including ANOVA main effects and interactions, not just post-hocs). Figure 2: Be explicit about the follow up test statistic for your Tukey post-hoc (either q/t/F value acceptable). Figure 4B and 5C require a 3 way interaction analysis.

2) Please provide histology figures with cannulae placements and viral expression spread for each subject, where appropriate.

3) Please revise the Introduction. References are missing, the argument is not constructed well, paragraphs do not fit with each other. The link between GluN2B and GluA2 and NOR-LC has not been made. A brief introduction to the procedure is necessary to go along with the Introduction.

4) Provide some discussion of the choice to use home cage controls, focusing on pros and cons.

[Editors' note: further revisions were suggested prior to acceptance, as described below.]

Thank you for resubmitting your work entitled "Noradrenergic projections from the locus coeruleus to the amygdala constrain fear memory reconsolidation" for further consideration by *eLife*. Your revised article has been evaluated by Kate Wassum (Senior Editor) and a Reviewing Editor.

The manuscript has been improved but there are some remaining issues that need to be addressed before acceptance, as outlined below:

Thank you for proving the tables with the statistical analyses. Some additional clarification is required in the text. In all instances an ANOVA is mentioned, followed by the Tukey post-hoc, followed by an F statistic. It is unclear what the F statistic refers to? Is it the ANOVA? An overall ANOVA? An interaction? Please check all instances where this is reported and clarify what the F statistic refers to by having it follow the appropriate name of the analysis it belongs to. It is acceptable to only report the interaction in some instances without the main effects as those are present in the table, but please specify. For example in paragraph two of subsection “Replicating the behavioral effects of 1 pairing vs. 10 pairings fear conditioning protocol”, the track changes show a deletion of the Tukey post-hoc between the ANOVA and F statistic, suggesting that the F statistic reported is of the ANOVA. This would be appropriate in all instances in the text. However, what is missing on is a reference as to what exactly the F statistic refers to, which presumably is the interaction. In addition, a repeated measures ANOVA is reported but this should be a mixed ANOVA (between group main effect of 1P vs. 10P; repeated measures of trials, and the interaction), this is also the case in paragraph three, so please check all instances.

Lastly, at the end of paragraph one of the Introduction a citation is needed as opposed to calling the figure.

---

## [Author Response]

Essential revisions:1) Please report all statistics (including ANOVA main effects and interactions, not just post-hocs). Figure 2: Be explicit about the follow up test statistic for your Tukey post-hoc (either q/t/F value acceptable). Figure 4B and 5C require a 3 way interaction analysis.

To fully address the reviewers call for statistical reporting, we have uploaded tables with all the statistics outputs as supplemental material. Also, we have made more explicit in the text the specific analyses and comparisons being described and included the t values for Tukey post-hoc. As suggested, Figure 4B and 5C were reanalyzed with a three-way ANOVA.

2) Please provide histology figures with cannulae placements and viral expression spread for each subject, where appropriate.

These figures are included for experiments 2.C, 4.B and 5.C.

3) Please revise the Introduction. References are missing, argument is not constructed well, paragraphs do not fit with each other. The link between GluN2B and GluA2 and NOR-LC has not been made. A brief introduction to the procedure is necessary to go along with the Introduction.

Thank you for pointing this out. We have included new references that were indeed missing and restructured some paragraphs to better establish the links. We have also expanded the last paragraph of the Introduction to explain the general procedures, and made several minor changes across the manuscript to improve its readability. We hope that this addresses the reviewers’ concerns.

4) Provide some discussion of the choice to use home cage controls, focusing on pros and cons.

In Figure 3 we were interested in evaluating how GluN2B and GluA2 change after 1P and 10P memory formation. To this end, we used home cage animals to see how these molecules are expressed when no learning occurs. However, other options for such control could have been used.

One option would be a no-shock control. One issue with this option would be that we would not have a single control for both 1P and 10P groups. Given that 1P and 10P training have different lengths, each training protocol would require a separate no-shock control. The no-shock control could be informative if considering the role of the contextual processing, and a structure such as the hippocampus. But it would be less valuable when studying the amygdala which is not a structure specialized in contextual processing.

Another option would be the use of unpaired control groups where animals would be exposed to the shocks but not to the tone presentations. The unpaired control would provide animals with no auditory fear learning but normal contextual learning. Again, this would require 2 control groups, one for each training protocol. Also, given that contextual fear was not the focus of this study, it would not add much for our goals.

The home cage control provides a clean way of assessing baseline levels of the targeted proteins. It gives us a single baseline value that can be compared with the data of both 1P and 10P groups. It does not address parameters such as context exposure and hippocampus-dependent learning, but on the other hand it avoids contamination intrinsic to other sorts of controls. Since our goal was to have a robust and comparable baseline value, we chose to use home cage animals. A sentence summarizing this was included in the first paragraph of subsection “Quantification of synaptic plasticity molecules between reconsolidation-permissive vs. resistant memories in the BLA”.

[Editors' note: further revisions were suggested prior to acceptance, as described below.]

Thank you for proving the tables with the statistical analyses. Some additional clarification is required in the text. In all instances an ANOVA is mentioned, followed by the Tukey post-hoc, followed by an F statistic. It is unclear what the F statistic refers to? Is it the ANOVA? An overall ANOVA? An interaction? Please check all instances where this is reported and clarify what the F statistic refers to by having it follow the appropriate name of the analysis it belongs to. It is acceptable to only report the interaction in some instances without the main effects as those are present in the table, but please specify. For example in paragraph two of subsection “Replicating the behavioral effects of 1 pairing vs. 10 pairings fear conditioning protocol”, the track changes show a deletion of the Tukey post-hoc between the ANOVA and F statistic, suggesting that the F statistic reported is of the ANOVA. This would be appropriate in all instances in the text. However, what is missing on is a reference as to what exactly the F statistic refers to, which presumably is the interaction. In addition, a repeated measures ANOVA is reported but this should be a mixed ANOVA (between group main effect of 1P vs. 10P; repeated measures of trials, and the interaction), this is also the case in paragraph three, so please check all instances.Lastly, at the end of paragraph one of the Introduction a citation is needed as opposed to calling the figure.

We have carefully checked the statistical reporting and made several changes across the Results section to improve its clarity. Most of the outputs of ANOVAs and Tukey’s post hoc tests are now presented separately, and all the specific comparisons are now clearly stated. When the statistical test is mentioned inside the parenthesis, it always appears to the left of its output. We believe that with these changes it is now clear which test and comparison the statistic values correspond to, even when many outputs are inside the same parenthesis.

Of note, in the extinction experiment mentioned by the reviewers, the ANOVA referred to a main effect of group, not to a group x session interaction which was not significant. This was expected since the 10P group exhibited higher freezing than the 1P group regardless of the session.

The reviewers are correct about the analysis reported as repeated measures ANOVAs – we did conduct a mixed ANOVA but wrongly called it a repeated measure. This was fixed.

Regarding the sentence lacking citation at the end of the first paragraph, it was removed and integrated into the second paragraph. It now contains both the required citations and the call for the figure. We have also added another reference to what is now the end of the first paragraph.